# Tissue Integration and Degradation of a Porous Collagen-Based Scaffold Used for Soft Tissue Augmentation

**DOI:** 10.3390/ma13102420

**Published:** 2020-05-25

**Authors:** Jordi Caballé-Serrano, Sophia Zhang, Anton Sculean, Alexandra Staehli, Dieter D. Bosshardt

**Affiliations:** 1Robert K. Schenk Laboratory of Oral Histology, School of Dental Medicine, University of Bern, 3010 Bern, Switzerland; jordicase@uic.es (J.C.-S.); sophia.zhang@zmk.unibe.ch (S.Z.); 2Department of Periodontology, School of Dental Medicine, University of Bern, 3010 Bern, Switzerland; anton.sculean@zmk.unibe.ch (A.S.); alexandra.staehli@zmk.unibe.ch (A.S.); 3Department of Oral and Maxillofacial Surgery, School of Dental Medicine, Universitat Internacional de Catalunya, 08017 Barcelona, Spain

**Keywords:** collagen, elastin, scaffold, biomaterial, tissue engineering, tissue response, wound healing, degradation, immunohistochemistry

## Abstract

Collagen-based scaffolds hold great potential for tissue engineering, since they closely mimic the extracellular matrix. We investigated tissue integration of an engineered porous collagen-elastin scaffold developed for soft tissue augmentation. After implantation in maxillary submucosal pouches in 6 canines, cell invasion (vimentin), extracellular matrix deposition (collagen type I) and scaffold degradation (cathepsin k, tartrate-resistant acid phosphatase (TRAP), CD86) were (immuno)-histochemically evaluated. Invasion of vimentin^+^ cells (scattered and blood vessels) and collagen type I deposition within the pores started at 7 days. At 15 and 30 days, vimentin^+^ cells were still numerous and collagen type I increasingly filled the pores. Scaffold degradation was characterized by collagen loss mainly occurring around 15 days, a time point when medium-sized multinucleated cells peaked at the scaffold margin with simultaneous labeling for cathepsin k, TRAP, and CD86. Elastin was more resistant to degradation and persisted up to 90 days in form of packages well-integrated in the newly formed soft connective tissue. In conclusion, this collagen-based scaffold maintained long-enough volume stability to allow an influx of blood vessels and vimentin^+^ fibroblasts producing collagen type I, that filled the scaffold pores before major biomaterial degradation and collapse occurred. Cathepsin k, TRAP and CD86 appear to be involved in scaffold degradation.

## 1. Introduction

Collagen is the most abundant extracellular matrix protein in the body. Besides having important structural functions, collagen supports cell attachment, cell differentiation, repair, and tissue regeneration [1,2]. Biomaterials in the form of collagen-based scaffolds hold great promise for both bone and soft tissue engineering to replace damaged or lost tissues, since these biomaterials provide an environment close to the native extracellular matrix [3,4,5]. Collagen-based scaffolds should at least provide: (1) high biocompatibility, (2) a highly porous structure with interconnected pores to allow influx of progenitor cells and blood vessels, (3) mechanical properties similar to the native tissue and (4) degradation properties and kinetics that match the characteristic speed of the tissue to be regenerated.

The mechanical and degradation properties of collagen-based scaffolds can be tuned by altering the ratio between collagen and elastin on one hand, and by the type and degree of chemical cross-linking on the other hand [6]. Collagen provides mechanical strength and stiffness, whereas elastin is responsible for elasticity. Cross-linking, in general, enhances both structural stability and resistance against degradation.

The autologous connective tissue graft from the palate is still the gold standard for soft tissue augmentations in the oral cavity, but because of additional time and morbidity, the search is still going on for the ideal biomaterial to replace the autologous graft. Lately, a collagen-based scaffold with interconnected pores from porcine origin was developed for soft tissue augmentation around teeth and dental implants [7]. Mechanical testing showed that the scaffold remained elastic for at least 14 days [8]. Although preclinical studies showed promising results in terms of the gain in tissue thickness, the biology behind it remains largely unknown. We previously demonstrated that this biomaterial elicits a short inflammatory phase followed by rapid cell proliferation and the fast influx of blood vessels and mesenchymal cells [9]. What occurs inside the scaffold pores, and how the scaffold degrades and becomes integrated in the host tissue, is still not known.

The aim of the present study was therefore to characterize the invading cells, to evaluate extracellular matrix deposition within the scaffold pores, and to assess scaffold degradation over a period of 90 days. To this end, a canine implantation model was used [9] and immunohistochemistry with antibodies against vimentin (for mesenchymal cells), collagen type I (for extracellular matrix formation), and cathepsin K, tartrate-resistant acid phosphatase (TRAP) and CD86 (for scaffold degradation) were applied. Furthermore, collagen type I deposition was histomorphometrically quantified to show the dynamics of extracellular matrix formation, both into the scaffold and over time.

## 2. Materials and Methods

The present study had been approved by NAMSA Ethical Committee (ID number: 0041573) and had been performed following the policies and principles of the Organization for Economic Cooperation and Development Good Laboratory Practice regulations and the European Union Guidelines (86/609/EEC).

Production of the collagen-based scaffold (Fibro-Gide^®^ prototype, Geistlich, Wolhusen, Switzerland) involved isolating, processing and milling two collagenous tissues, followed by mixing the materials at a 3.5–4.5:0.5–1.5 ratio in an aqueous slurry. After freeze-drying, the resulting composite was treated with a crosslinking agent, washed with water, freeze-dried again, and terminally sterilized by gamma irradiation. The detailed production process is proprietary. The final product was composed of 60%–96% (w/w) porcine collagen type I and III and 4%–40% elastin. Furthermore, the average pore diameter was 92 µm and the volume porosity was 93% with interconnected pores, as determined by mercury intrusion porosimetry [8]. Chemical cross-linking was used to increase the stiffness of the scaffold. The critical mechanical properties were described by retention hysteresis under repetitive compressive load of 12.1 kPa, whereby the first order retention was less than 45% after 49 cycles. This equated to a thickness retention after successive loading cycles of more than 85%. The measurements were carried out in saline solution at 37 °C.

### 2.1. Surgical Phase

A concise description of the protocol concerning the animal selection and surgical phase is described in a recent publication [10]. Briefly, in six mature dogs with a weight between 11.6 and 14.5 kg, the maxillary premolars were atraumatically extracted to create one edentulous space on each side of the maxilla. After 90 days, a full thickness flap was elevated and a collagen-based scaffold (Fibro-Gide^®^ prototype, Geistlich, Wolhusen, Switzerland), measuring 6 mm × 6 mm × 10 mm, was placed between the bone and the repositioned buccal flap. The wounds were closed by primary intention. Sacrifice of animals was performed after 4 h and at days 4, 7, 15, 30 and 90.

### 2.2. Histologic Processing and Descriptive Analysis

After resection, the maxillary samples were immediately immersed in 10% buffered formalin. Upon arrival in the histology laboratory, the tissue samples were trimmed and immersed in 4% formalin for 10 days. To perform demineralization, tissue samples were immersed in 10% ethylenediaminetetraacetic acid (EDTA) for 25 days. Samples were divided in thirds and embedded in paraffin, methylmethacrylate (MMA) or in acrylic resin (LR White, Sigma-Aldrich, Saint Louis, MO, USA). Paraffin sections were cut at 5 μm, while MMA-embedded tissues were cut with a diamond-coated disc and ground to a thickness of 100 μm. LR White sections were cut at 1 μm with a diamond knife on a Reichert Ultracut E microtome (Leica Microsystems, Wetzlar, Germany). MMA-embedded sections were stained with basic fuchsin and toluidine blue and LR White-embedded sections were double stained with toluidine blue and basic fuchsin. Tissue sections embedded in paraffin were stained with hematoxylin/eosin and Giemsa. Immunohistochemical staining with antibodies that target proteins expressed in mesenchymal cells (Vimentin), in M2 phenotypes of the macrophage cell lineage (CD86), in the newly formed extracellular matrix (collagen type I), and in proteases involved in extracellular matrix degradation (Cathepsin K) was applied. In addition, histochemical staining for tartrate-resistant acid phosphatase (TRAP), an enzyme expressed during mineralized tissue degradation and in soft tissue pathologies, was performed. Micrographs were taken using a digital camera (AxioCam MRc, Carl Zeiss, Oberkochen, Germany), connected to a microscope (Axio Imager M2, Carl Zeiss, Oberkochen, Germany). Observations were made in two regions of the scaffold; at the periphery next to the bone and at the center of the scaffold.

### 2.3. Preparation for Vimentin Immunohistochemistry

Sections embedded in paraffin underwent immunohistochemical staining with an anti-vimentin antibody (clone V9, Vimentin, Sigma, Saint Louis, MO, USA). Blocking with defatted milk was performed for 30 min and Dako EnVisionTM + Dual Link System-HRP (DAB+) was used. Incubation with the antibody was during 2 h at room temperature. The antibody was diluted 1:2000. Samples were counterstained using Mayer’s hematoxylin solution (Merck, Darmstadt, Germany).

### 2.4. Preparation for Collagen Type I Immunohistochemistry

Sections embedded in paraffin underwent immunohistochemical staining with an anti-collagen type 1 antibody (clone 6308, Collagen 1, Abcam, Cambridge, UK). Heat-induced epitope retrieval was performed at 80–82 °C for 10 min with a citrate solution. Blocking with defatted milk was performed for 30 min and Dako EnVisionTM + Dual Link System-HRP (DAB+) was used. Incubation with the antibody was done for 2 h at room temperature. The antibody was diluted 1:100. Samples were counterstained using Mayer’s hematoxylin solution (Merck, Darmstadt, Germany).

### 2.5. Quantitative Analysis of Collagen Type I

For the quantitative analysis of collagen type I, 3 animals were used (7, 15, and 30 days of healing), because collagen deposition in the scaffold was not observed at 4 h and at 4 days, and 90 days was not of interest because the active phase of collagen synthesis and deposition was over and replaced by tissue remodeling. The area fractions of labeled collagen type I were determined in high-resolution images in 5 rectangles per section and region (i.e., 5 rectangles in the periphery and 5 sections in the center of the collagen-based scaffold). The rectangle size was 0.5 mm × 0.75 mm (area = 0.375 mm^2^). The central rectangles were placed immediately adjacent to the peripheral rectangles. The rectangles in the periphery were all evenly distributed on the bone side with similar distances between them, the same was true for the central rectangles. The Zeiss Zen software (Zeiss, Oberkochen, Germany) was used to capture the DAB-stained collagen type I and to convert the values into percentages: (labeled area/total area) × 100.

### 2.6. Preparation for Giemsa Histochemistry

Paraffin-embedded tissue sections were treated with Giemsa staining using Giemsa stock-solution (T862.2, Carl Roth GmbH, Karlsruhe, Germany), as recommended by the manufacturer.

### 2.7. Preparation for Tartrate-Resistant Alkaline Phosphatase (TRAP) Histochemistry

Sections embedded in paraffin underwent tartrate-resistant acid phosphatase (TRAP) histochemical staining. Sections were treated with azo staining using naphthol AS-TR phosphate (Fluka, Sigma-Aldrich, Saint Louis, MO, USA), coupled with fast red violet TR salt (Chroma, Stuttgart, Germany).

### 2.8. Preparation for Cathepsin K Immunohistochemistry

Sections embedded in paraffin underwent immunohistochemical staining with an anti-cathepsin K antibody (clone 19027, cathepsin K, Abcam, Cambridge, UK). Blocking with defatted milk was performed for 30 min and Dako EnVisionTM + Dual Link System-HRP (DAB+) was used. Incubation with the antibody was done for 1.5 h at room temperature. The antibody was diluted 1:500. Samples were counterstained using Mayer’s hematoxylin solution (Merck, Darmstadt, Germany).

### 2.9. Preparation for CD86 (B7-2) Immunohistochemistry

Sections embedded in paraffin underwent immunohistochemical staining with an anti-CD86 antibody (clone EP1158Y, CD86, Abcam, Cambridge, UK). Heat-induced epitope retrieval was performed at 85 °C for 10 min with a citrate solution. Blocking with defatted milk was performed for 30 min and Dako EnVisionTM + Dual Link System-HRP (DAB+) was used. Incubation with the antibody was done for 2 h at room temperature. The antibody was diluted 1:100. Samples were counterstained using Mayer’s hematoxylin solution (Merck, Darmstadt, Germany).

## 3. Results

Healing of the animals did not present complications. Tissue samples were collected and processed for histological analysis according to the Material and Methods section. Wound healing outside the biomaterial was normal, as reported recently [9]. Histological inspection revealed that the porous collagen-based scaffold (Figure 1A) was present close to the bone surface at all healing periods (Figure 1B). The scaffold presented a large honeycomb-like interconnected porous structure (Figure 1A), consisting of an amorphous, sheet-like matrix (i.e., the collagenous part of the scaffold), with embedded, rod-like structures (i.e., the elastin part of the scaffold) (Figure 1C) and pores that were rapidly filled with fluid and blood cells (Figure 1C). At 90 days, residual scaffold material was still visible. It consisted of islands of the rod-like scaffold part, that were dispersed among, and well-integrated in, the surrounding host soft connective tissue (Figure 1D). For reasons of consistency and standardization, the following description of the results will be confined to the region facing the bone surface.

### 3.1. H&E Staining

Up to day 7, the central part of the scaffold showed a clear delay in the invasion of blood vessels and mesenchymal-like cells and extracellular matrix formation when compared to the peripheral part. Immediately after implantation, the pores of the scaffold were mainly filled with erythrocytes and blood plasma/wound fluid (Figure 2A,G). At 4 days, a fibrin network with trapped leukocytes was present within most scaffold pores (Figure 2B,H) and small blood vessels were seen in the soft connective tissue next to the scaffold (Figure 2B). Very few blood vessels and mesenchymal-like cells were seen in the outermost portion of the scaffold. At 7 days, blood vessels gradually grew in the scaffold pores and increased in size and number at the periphery, and the number of invading mesenchymal-like cells increased as well (Figure 2C). Furthermore, the pores became filled with an extracellular matrix (Figure 2C). Compared to the periphery (Figure 2C), this development was less advanced in the central part of the scaffold (Figure 2I). After 15 days, more blood vessels were seen in the scaffold pores, the mesenchymal-like cells inside the pores had increased in quantities, and extracellular matrix fill was more advanced (Figure 2D,K). At 30 days, many blood vessels, mesenchymal-like cells now resembling thin fibroblasts, and an extracellular matrix, filled the scaffold pores both in the peripheral (Figure 2E) and central (Figure 2L) part of the scaffold. At 90 days, packages of condensed residual scaffold elastin were fully integrated in the host soft connective tissue (Figure 2F,M).

### 3.2. Immunohistochemistry with the Anti-Vimentin Antibody

Vimentin labeling in the scaffold pores was observed, both in scattered cells and blood vessels. Vimentin+ scattered cells and blood vessels were located in the connective soft tissue next to the scaffold at 4 days (Figure 3A). No labeled cells were seen in the peripheral scaffold pores at 4 days (Figure 3A) and in the central part of the scaffold at 4 (Figure 3F) and 7 days (Figure 3G). However, many vimentin+ scattered cells and blood vessels were seen in the peripheral scaffold pores at 7 days (Figure 3B). At 15 days, many vimentin+ scattered cells and blood vessels occupied the scaffold pores (Figure 3C,H), and more vimentin+ cells were observed in the periphery (Figure 3C) than in the central part (Figure 3H). At 30 days, most of the vimentin+ cells were located in the center of the scaffold (Figure 3I), while some vimentin positive cells could still be found in the peripheral pores and in the external host tissue (Figure 3D). At 90 days, vimentin+ cells were mainly distributed in the host connective tissue surrounding the residual scaffold packages, with no differences between peripheral and central regions (Figure 3E,K). The negative controls did not show positive labeling for vimentin (data not shown).

### 3.3. Immunohistochemistry with the Anti-Collagen Type I Antibody

Inspection of the not injured soft connective tissue part of the oral mucosa and of bone revealed positive immunostaining for collagen type I. Immunostaining for collagen type I inside the biomaterial scaffold was first observed after 7 days. Collagen type I labeling progressively increased from the periphery to the central part of the scaffold and the area occupied with labeled collagen type I increased from 7 to 30 days (Figure 4). The labeling intensity was always stronger for the newly formed collagen inside the scaffold than for collagen in the surrounding soft connective tissue and in bone. There was no collagen type I immunostaining observed at all at 4 days in the periphery (Figure 4A) and at 4 (Figure 4E) and 7 days (Figure 4F) in the central part of the scaffold. The first collagen type I immunostaining in the scaffold pores was seen at day 7 in the peripheral part (Figure 4B). At 15 days, dense and compact structures positive for collagen type I were filling the pores of the scaffold at the periphery (Figure 4C). The first collagen type I immunostaining in the center of the scaffold was seen at 15 days (Figure 4G), but it was less dense than that in the periphery (Figure 4C). At 30 days, the extracellular matrix positive for collagen type I was more dense and almost completely filled the scaffold pores irrespective of their location (Figure 4D,H). At 90 days, the extracellular matrix surrounding the residual scaffold occupied a large tissue portion and was positive for collagen type I (not shown). At all observation periods where positive labeling for collagen type I was observed, elongated and slender cells (fibroblasts) were seen intermingling with the newly formed collagen matrix. Negative controls did not show positive labeling for collagen type I (data not shown).

### 3.4. Quantitative Histological Analysis for Collagen Type I

Quantification of collagen type I immunostaining showed an increase in the area occupied by immunostaining from 7 to 30 days, both in the periphery and in the center of the scaffold (Figure 5). For days 7 and 15, the areas of collagen labeling were clearly larger in the periphery than in the center. At later time points, this difference was no more visible. In the center, the immunostained area significantly increased from day 7 to day 15 and from day 15 to day 30.

### 3.5. Degradation of the Scaffold

Resin sections stained with toluidine blue and basic fuchsin differentially stained the collagenous and elastin parts of the scaffold and allowed therefore to study the alterations these two constituents underwent over time (Figure 6). The collagenous part was stained in dark purple, while the elastin part was stained in dark pink (Figure 6). At early time points, collagen and elastin of the scaffold clearly stood out (Figure 6A–C). Later, the collagen part disappeared, leaving mainly elastin embedded in newly formed collagen that stained bluish (Figure 6D,E). The light blue or purple color around the elastin is newly formed collagen. At 90 days, only packages of elastin fibers were left and integrated into the newly formed soft connective tissue (Figure 6F).

To evaluate the cell-mediated degradation of the scaffold, we studied the appearance of multinucleated cells using Giemsa and TRAP histochemistry together with cathepsin K and CD86 immunostaining (Figure 7). Giemsa staining gave a differential staining of multinucleated cells, so that they stood out against the surrounding tissue elements (Figure 7A). Most multinucleated cells were observed 15 days after scaffold implantation in the peripheral part of the scaffold (Figure 7A). No multinucleated cells were detected at 4 h, and at 4 and 90 days; a few only at 7 and 30 days. These multinucleated cells were medium-sized and in close contact to the scaffold surface. In the same region as the image shown in Figure 7A, adjacent consecutive sections revealed cells of similar size that were positive for cathepsin K (Figure 7B), TRAP (Figure 7C), and CD86 (Figure 7D). Before and after 15 days and in the center of the scaffold, no multinucleated cells positive for cathepsin K or CD86 were found. Very few TRAP-positive cells remained in the scaffold up to day 30, mainly at the periphery (data not shown). Negative controls did not show positive labeling for TRAP, cathepsin K or CD86 (data not shown).

## 4. Discussion

The idea behind using biomaterials for tissue regeneration is to avoid the drawbacks of autogenous tissue harvesting and transplantation [11]. Another advantage of biomaterials is their availability in large amounts. Collagen is the most commonly used biomaterial to augment or regenerate soft tissues, since collagen scaffolds mimic very well the natural extracellular matrix environment. Recently, we have shown that the collagen-based scaffold tested in the present study preserved its porous structure after implantation and presented a short inflammatory phase followed by rapid influx of blood vessels and mesenchymal-like cells undergoing rapid cell proliferation [9]. The aim of the present study was to characterize the invading cells, to evaluate extracellular matrix deposition within the scaffold pores, and to assess scaffold degradation over a period of 90 days.

Our results demonstrate that vimentin-positive cells progressively invaded the pores of the collagen-based scaffold, starting at day 7 in the periphery and reaching the center at day 15. The highest numbers of vimentin-positive cells in the pores were seen at day 15 in the periphery and the center of the scaffold. The vimentin-positive cells were either scattered or associated with blood vessels. The scattered cells became increasingly slender and elongated and were aligned parallel to the fibrous extracellular matrix filling the scaffold pores. Interestingly, vimentin labeling was also associated with the invading blood vessels. Vimentin, an intermediate cytoskeletal filament, is mainly expressed in cells of mesenchymal origin [12] and is often used as a fibroblast marker. Vimentin is also an important cytoskeletal filament in endothelial cells and plays a role in cell polarity, migration and vascularization [12,13], as well as wound healing by controlling fibroblast proliferation [14]. Collectively, our data support that blood vessels, together with fibroblasts, invaded the scaffold pores.

Our immunohistochemical and histomorphometric data show that the pores of the collagen-based scaffold became increasingly filled with collagen type I, starting at day 7 from the periphery and progressing towards the center. At day 30, most of the pores were considerably filled with newly formed collagen. The collagen type I labeling, together with the vimentin labeling, strengthens the hypothesis that the invading cells are fibroblasts. Newly formed collagen type I can be specifically stained using antibodies [15] or using histochemical stains such as Masson’s trichrome to study the newly synthesized collagen fibrils, as shown in a recent publication where an elastin scaffold was used for tissue engineering purposes of cardiovascular disorders [16]. Very recently, it was shown that the physical characteristics of biomaterial scaffolds modulate macrophage response [17] and polarization [18]. Macrophages are involved in tissue repair by regulating vascularization [19,20], fibroblast proliferation and activity [21], and the rapid synthesis of collagen fibrils [22]. These data are in line with our previous data, showing a burst of macrophages within the scaffold pores and their rapid disappearance, as well as fast cell proliferation [9], and with the results from the present study demonstrating fast ingrowth of vimentin-positive fibroblasts after the influx of macrophages, followed by rapid collagen synthesis.

Recently, it was suggested that the scaffold tested undergoes degradation and that the collagenous part degrades faster than the elastin fibers eventually leaving elastin fibers for a longer period of time within the tissue [9]. Here, we show now that markers associated with matrix degradation were almost exclusively expressed at 15 days after scaffold implantation. The expression of cathepsin K, TRAP, and CD86 was mainly found in medium-sized multinucleated cells at the interface between scaffold and neighboring connective soft tissue. Cathepsin K, a member of the cysteine proteases, has a high matrix-degrading activity, especially for collagen type I [23,24]. Cathepsin K was initially thought to be exclusively expressed in osteoclasts and therefore used as one of the cell markers for osteoclasts. Cathepsin K expression, however, was also observed in epithelioid cells and multinucleated giant cells in soft tissues, suggesting that these cells also possess proteolytic capability in non-mineralizing tissues [17]. Multinucleated giant cells strongly express CD86 [25,26]. Thus, our results of CD86-positive multinucleated cells suggest an involvement of these cells in the degradation of the collagen-based scaffold. Co-localization of cathepsin K and CD86 has been described in macrophages related to degradation and rupture of carotid plaques [27], but not in soft tissue biomaterials so far. Other studies have already pointed to macrophages playing a role in biomaterial degradation [28,29]. Tartrate-resistant acid phosphatase (TRAP), a proteolytic enzyme involved in bone matrix degradation, was originally used as an osteoclast-specific marker. However, TRAP is now considered as a widespread molecule with functions in hard and soft tissues and is expressed by osteoclasts, macrophages, dendritic cells, and other cell types [30]. To the best of our knowledge, there are no other data in the literature showing expression of TRAP in association with collagen degradation in soft tissues. Studies analyzing degradation of collagen-based biomaterials have found CD86-positive cells within the biomaterial, but no TRAP staining [31]. Since the collagenous part of the investigated scaffold underwent degradation, it is very likely that TRAP was involved in this degradation process. Thus far, TRAP was mainly associated with the degradation of the organic bone matrix. Taken together, our data suggest that medium-sized multinucleated cells expressing at least cathepsin K, TRAP, and CD86 play a role in the degradation of collagenous scaffolds. No data reporting on the co-localization of these three markers in macrophages were found in the literature. It is very possible that cathepsin K and TRAP are involved in the degradation of the collagenous part of the scaffold, since up to day 7, the elastin fibers remained embedded in the collagen matrix, whereas from day 15 on, elastin fibers appeared to be released from the collagen-elastin scaffold (see Figure 6). At 90 days, only packages of elastin fibers were seen embedded in newly formed host collagen fibers. A challenging task in biomaterial development for soft tissue augmentation is to match the degradation time of the biomaterial with the extracellular matrix formation dynamics, to avoid early biomaterial collapse. The present data indicate that this balance is successfully achieved. A good ratio between collagen and elastin together with collagen cross-linking may be responsible for this.

Like other studies, the present study also has limitations. One limitation is the small sample size due to the study design, which aimed at studying the sequence of events over a period of 90 days, using 6 observation periods. The study may thus be considered as a pilot experiment and the data as preliminary. Furthermore, only a negative control was used, i.e., sham-operated sites. Future experiments may involve a larger number of tissue samples and other biomaterials, for comparative reasons. Nevertheless, the study provides novel and unique data of clinical relevance and for future studies in the biomaterial field.

## 5. Conclusions

Taken together, the results from the present and a recent study [9] demonstrate that the porous scaffold elicits a short inflammatory phase, maintains long-enough volume stability to allow influx of blood vessels and vimentin-positive fibroblasts producing collagen type I that fills the scaffold pores, before major biomaterial degradation and collapse occurs. These histologic data form the rational for the clinical application of this biomaterial for soft tissue augmentation around dental implants and teeth. It needs to be determined whether this collagen-based scaffold can also be used as a tissue engineering biomaterial for periodontal regeneration and the restoration of bone defects.

## Figures and Tables

**Figure 1 materials-13-02420-f001:**
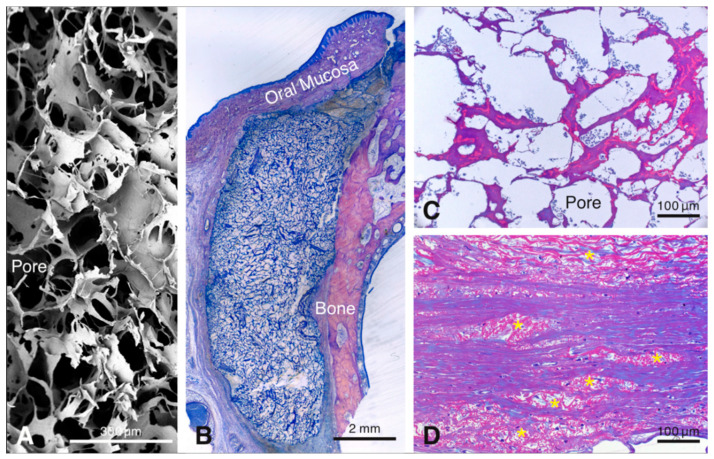
Scanning electron microscopic image illustrating the three-dimensional-structure of the scaffold (**A**). Overview of a methylmethacrylate (MMA)-embedded tissue section showing the scaffold between the bone and the soft connective tissue 4 days after implantation (**B**). The resin section illustrates the pores of the scaffold filled with blood plasma and erythrocytes after 4 days (**C**). Resin section 90 days after implantation of the scaffold showing elastin fiber packages integrated in the host tissues (**D**).

**Figure 2 materials-13-02420-f002:**
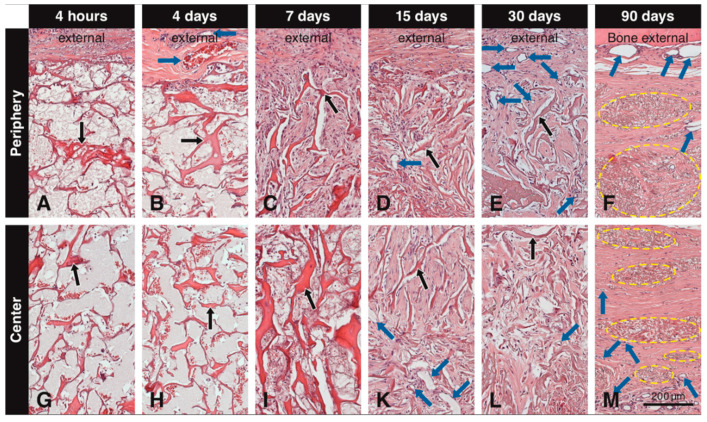
Tissue sections embedded in paraffin and stained with H&E showing the scaffold (black arrows) at different time points at the periphery and the center. Immediately after implantation and after 4 days, the pores of the scaffold are mainly filled with blood plasma/wound fluid and erythrocytes (**A**,**B**,**G**,**H**). At 4 days, blood vessels (blue arrows) can be seen adjacent to the scaffold, but not penetrating it (**B**). At 7 days, cells start to invade the scaffold from the periphery and initial filling of the pores with extracellular matrix begins (**C**), while these cells do not reach the central portion yet (**I**). At 15 days, blood vessels are clearly seen in the scaffold pores both at the periphery (**D**) and in the center (**K**), while mesenchymal-like cells inside the pores have increased in amounts, and extracellular matrix fill is more advanced (**D**,**K**). At 30 days, blood vessels increase in number at the periphery (**E**) and in the center of the scaffold (**L**) and newly formed tissue within the pores presents a compact structure throughout the scaffold (**E**,**L**). At 90 days, dense packages of residual elastin (yellow circles) are present at the periphery and the center of the scaffold, along with large blood vessels external to the scaffold and smaller blood vessels in the newly formed soft connective tissue around the residual elastin (**F**,**M**).

**Figure 3 materials-13-02420-f003:**
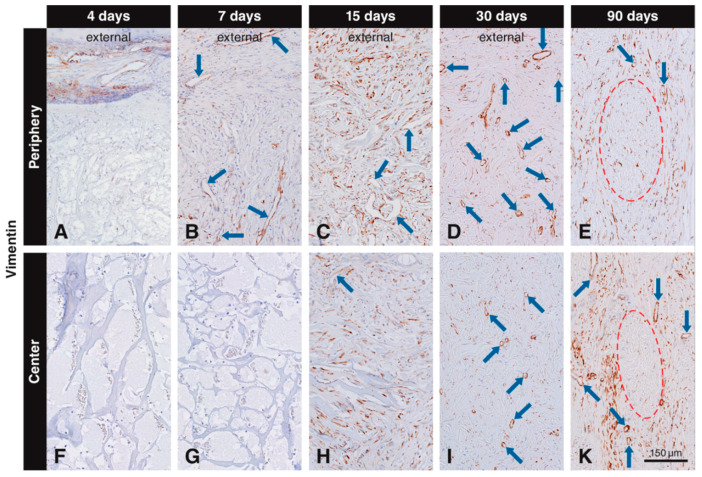
Immunohistochemical staining for vimentin to label mesenchymal cells. Brown (DAB^+^) staining indicates vimentin^+^ mesenchymal cells. At four days, vimentin^+^ cells are located in the soft connective tissue around the collagen-based scaffold, but have not yet invaded it (**A**,**F**). At day 7, vimentin^+^ cells have invaded the periphery of the scaffold as scattered cells and blood vessels (blue arrows), but have not yet reached the center (**B**,**G**). At 15 days, numerous scattered cells and blood vessels positive for vimentin occupy the scaffold pores (**C**,**H**). At 30 days, most of the vimentin^+^ cells are located in the center of the scaffold (**I**), while some vimentin^+^ cells can still be found in the peripheral pores and in the external host tissue (**D**). Labeled cells with an elongated shape can be seen 90 days after implantation in the in the soft connective tissue surrounding the residual scaffold, both at the periphery (**E**) and in the center (**K**).

**Figure 4 materials-13-02420-f004:**
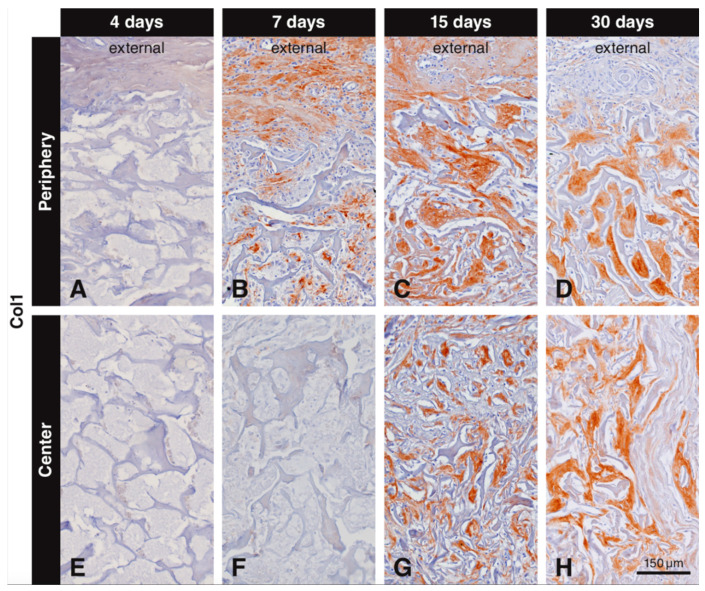
Immunohistochemical staining for collagen type I to label newly formed collagen within the scaffold pores. Collagen type I is stained in brown (DAB^+^). No collagen type I immunostaining is observed at 4 days (**A**,**E**), or at 7 days in the center of the scaffold (**F**). Collagen type I starts to fill the scaffold pores at 7 days starting from the periphery (**B**). At 15 days, compact structures positive for collagen type I fill the pores at the periphery of the scaffold (**C**) and start filling the central pores (**G**). At 30 days, newly formed collagen type I fills the scaffold pores, irrespective of their location (**D**,**H**).

**Figure 5 materials-13-02420-f005:**
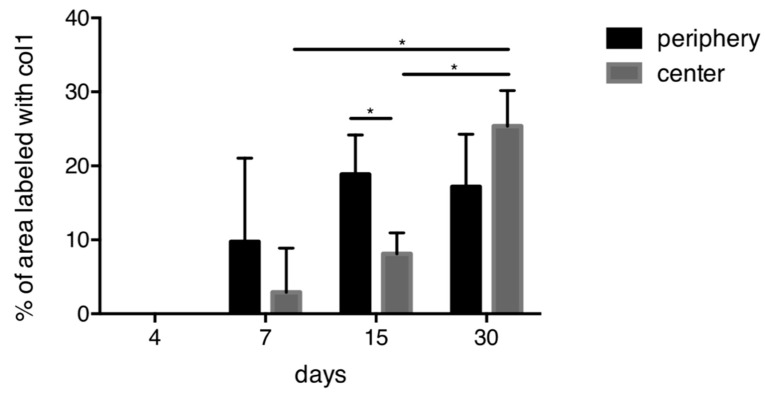
Quantitative analysis for collagen type I. An overall increase of collagen is observed over time. At 15 days, significantly more collagen is present in the periphery compared to the center of the scaffold. In the central part of the scaffold, significantly more collagen is present between 7 and 15 days as well as between 15 and 30 days. * *p* < 0.05.

**Figure 6 materials-13-02420-f006:**
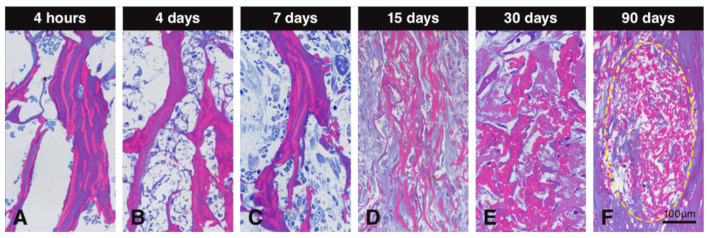
Resin sections stained with toluidine blue and basic fuchsin to evaluate degradation of the collagen-based scaffold. Collagen in the scaffold is stained in dark purple and elastin is stained in dark pink. The collagen content decreases over time, leaving elastin fibers (**A**–**F**). At early time points, collagen and elastin remain stable (**A**–**C**), while at later observation periods, the scaffold seems to become devoid of collagen. At 15 and 30 days, the collagenous constituents of the scaffold is mainly degraded, leaving only elastin fibers surrounded by newly formed collagen stained in light blue or purple (**D**,**E**). At 90 days, packages of elastin fibers (yellow circle) are fully integrated within the newly formed soft connective tissue (**F**).

**Figure 7 materials-13-02420-f007:**
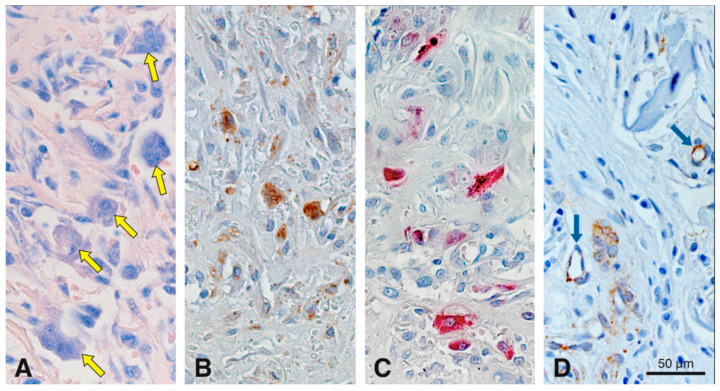
Histochemical staining for Giemsa and TRAP, and immunohistochemical stainings for cathepsin K and CD86 in serial sections to assess cell-mediated degradation of the scaffold at 15 days in precisely the same region. Medium-sized multinucleated cells (yellow arrows) are differentially stained with Giemsa (**A**). Cathepsin K (**B**), TRAP (**C**) and CD86 (**D**) stainings reveal that these cells are in direct contact with the collagen-based scaffold. Blue arrows in (**D**) points blood vessels partially positive for CD86. Cathepsin K^+^ (**B**) and CD86^+^ (**D**) cells, are stained in brown (DAB^+^).

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
