# Peer review of "Tissue Integration and Degradation of a Porous Collagen-Based Scaffold Used for Soft Tissue Augmentation"

_materials, 2020, doi:10.3390/ma13102420_

Round 1
Reviewer 1 Report
This manuscript presents data on the integration and degradation of collagen-elastin scaffolds in a maxillary submucosal canine model. Although it is scientifically relevant and results are well presented, my main objection regarding this manuscript is its overly high similarity to the previously published paper from the same group: doi:10.3390/ma12223721 (reference 9 on this manuscript).
As the authors state on lines 330-333, a lot of work has already been done in the previous paper. If we take a look, we will notice that figure 1 from this manuscript is almost identical to figure 1 from the previous paper. Figure 2 from both papers is also practically the same, the only difference is in the staining, in one H&E and in the other a trichrome.
The authors justify that in this current manuscript their objective is to characterize invading cells, evaluate ECM deposition and assess scaffold degradation. However, these were also already partially performed in the previous manuscript. To characterize the cells, here they use vimentin, cathepsin k, TRAP and CD86 as markers. In the previous study, they used PCNA, MAC387, TGM2, and CD86 (notice that CD86 is in both papers). Notions about the scaffold degradation were already present in the previous study since several stainings were already performed at 90 days. The same goes for ECM deposition. Although at this current manuscript immunohistochemistry for collagen 1 was well performed, the trichrome staining in the previous manuscript already gave insights about ECM deposition.
In conclusion, it is normal and acceptable to have follow-up studies, with different time-points of analysis, or different animal models. However, in this case, almost everything is the same, the same time-points, the same animal model, the same biomaterial, with the differences being only some immunohistochemistries and stainings. It is particularly surprising that the previous manuscript was very recently published (2 months ago), so that is unclear why the authors did not include the current data in the previous manuscript. Finally, although this current manuscript is very well written, it is scientifically interesting, and the experiments were well designed and performed, the high lack of originality, especially compared to a very recent paper doi.org/10.3390/ma12223721 from the same group, makes me suggest its rejection. As a final suggestion to the authors, if they want to include these current data in a novel manuscript, they could perform extensive more in-depth analysis on ECM deposition (analyze other markers), or focus deeply on the inflammatory aspect of the biomaterial, to try to separate it more clearly from their previous manuscript.
Author Response
Dear Reviewer 1
Thank you for the time you have spent on the review. Your insightful, constructive comments are going to improve our manuscript. Any changes in the revised manuscript are in track changes and indicate the location (page and lines) in the new version of the manuscript. Bellow you will find our point-by-point responses.
Reviewer 1: This manuscript presents data on the integration and degradation of collagen-elastin scaffolds in a maxillary submucosal canine model. Although it is scientifically relevant and results are well presented, my main objection regarding this manuscript is its overly high similarity to the previously published paper from the same group: doi:10.3390/ma12223721 (reference 9 on this manuscript).
Response to Reviewer 1: Thank you for your positive comments on the scientific relevance and presentation of our data. Regarding the similarity to a previously published paper, it is clear that there must be some similarity, since the data were generated from the same raw material originating from the same animal experiment, meaning that the same number of animals and the same healing periods were used. In our cover letter to the special Guest Editor Prof. Stylianou, we have clearly stated that the submitted manuscript is a continuation of a recently published paper in MATERIALS (Caballé-Serrano et al. Materials 2019,12,3721). However, the data are new, original, and scientifically relevant. All micrographs are new. Regarding the CD86 labeling, it is obvious why we have included one high-mag micrograph. You will find the explanation for this further below.
Reviewer 1: As the authors state on lines 330-333, a lot of work has already been done in the previous paper. If we take a look, we will notice that figure 1 from this manuscript is almost identical to figure 1 from the previous paper. Figure 2 from both papers is also practically the same, the only difference is in the staining, in one H&E and in the other a trichrome.
Response to Reviewer 1: We state on p. 11, lines 453-455, that 1) the implanted biomaterial elicited a short inflammatory phase; 2) rapid influx of blood vessels occurred; 3) rapid cell proliferation was analyzed; and 4) invasion of mesenchymal-like cells was observed. Inflammation was not assessed in the manuscript under revision. Ingrowth of blood vessels was not assessed in the manuscript under revision. Cell proliferation was also not assessed in the manuscript under revision. Invading cells were only characterized by morphology but not immunohistochemically. In the manuscript under revision, the immunohistochemical data on vimentin, collagen type 1, TRAP, and cathepsin k are all new. Regarding the similarity of figures 1 and 2, it is clear that they cannot be totally different from those of the first, published part, since it is the same study design and the same biomaterial. However, all picture are new and have previously not been published. If the Chief Editor wants these two figures to be omitted, there is no problem to do so, but it would be a pity, since they give a very good structural overview for the readers who are not familiar with this scaffold and they summarize what happened over time.
Reviewer 1: The authors justify that in this current manuscript their objective is to characterize invading cells, evaluate ECM deposition and assess scaffold degradation. However, these were also already partially performed in the previous manuscript. To characterize the cells, here they use vimentin, cathepsin k, TRAP and CD86 as markers. In the previous study, they used PCNA, MAC387, TGM2, and CD86 (notice that CD86 is in both papers). Notions about the scaffold degradation were already present in the previous study since several stainings were already performed at 90 days. The same goes for ECM deposition. Although at this current manuscript immunohistochemistry for collagen 1 was well performed, the trichrome staining in the previous manuscript already gave insights about ECM deposition.
Response to Reviewer 1: Thank you. The aim of the first paper was to assess the tissue response to a biomaterial. To achieve this goal, we have applied appropriate markers, i.e. an antibodies against macrophages, TGM2 for blood vessels, PCNA for cell proliferation, and CD86 for multinucleated giant cells. In the present study, vimentin immunohistochemistry (IHC) was used to characterize invading cells, collagen type 1 IHC was used to evaluate extracellular matrix formation, cathepsin k and TRAP IHC were used to study biomaterial degradation, and CD86 was used to further characterize the cells involved in the peak of matrix degradation observed at 15 days. It is true that we have been using CD86 IHC in the previous publication but not in the context of matrix degradation. Matrix degradation was an issue in the present study. In our opinion, it is justified to use just one single high-mag picture from an IHC incubation already used in a previous paper. The micrograph used has not been published previously and is only used to shed some more light on biomaterial degradation. The Masson’s trichrome staining in the previous paper indeed suggested that it could be collagen, but it was not clear. We have therefore used an antibody against collagen type 1 to prove this.
Reviewer 1: In conclusion, it is normal and acceptable to have follow-up studies, with different time-points of analysis, or different animal models. However, in this case, almost everything is the same, the same time-points, the same animal model, the same biomaterial, with the differences being only some immunohistochemistries and stainings. It is particularly surprising that the previous manuscript was very recently published (2 months ago), so that is unclear why the authors did not include the current data in the previous manuscript. Finally, although this current manuscript is very well written, it is scientifically interesting, and the experiments were well designed and performed, the high lack of originality, especially compared to a very recent paper doi.org/10.3390/ma12223721 from the same group, makes me suggest its rejection. As a final suggestion to the authors, if they want to include these current data in a novel manuscript, they could perform extensive more in-depth analysis on ECM deposition (analyze other markers), or focus deeply on the inflammatory aspect of the biomaterial, to try to separate it more clearly from their previous manuscript.
Response to Reviewer 1: Thank you for supporting our opinion that it is normal to do follow-up studies. The follow-up here was not the creation of a new experiment with killing more animals. The follow-up here was to use existing biopsy material but with another aim than in our previous publication. Since the material is the same as in the previous paper, number of animals, defect type, and healing periods must be the same. Yes, the difference was the use of other markers, markers appropriate to study tissue integration and degradation, an issue not covered in our first publication. After submission of the first paper, many people in our lab have been hard working to get the new data as soon as possible, since we thought that these new data are of high scientific value. What the outcome will be of the second part was not known at the time we submitted the first part.
Reviewer 2 Report
The work deals with the analysis of tissue integration of a porous collagen-elastin scaffold. It is mostly based on immuno-histochemical evaluation and presents original results. Still, major problems need to be adressed before acceptance for publication.
The fabrication of the scaffolds is not detailed, if the reason is industrial secret this should be explained and the main general characteristics should be reported (the composition is only briefly presented in the introduction), including the size of the used scaffolds, which is not even mentioned.
Along the same line, the authors claim in the discussion that the physical characteristics of biomaterial scaffolds modulate macrophage response and other host reactions, and that TRAP expression is reported for the first time, but no explanation of the obtained results and comparison based on the characteristics of the studied scaffold is proposed.
This leads to another flaw of the work, namely the absence of any control to test hypotheses based on the specific features of the studied scaffold. One could imagine that 3 out of the 6 dogs could have undergone implantation with another type of scaffold, or that the implantation could have been assymetric with the use of different types of scaffold on each side of the maxilla. If these experiments are not possible anymore, the discussion should at least explain why this was not performed and reflect real scientific questioning on the observations made, based on other results from the literature for example.
The results are not quantatively assessed and one might really wonder why 6 dogs were needed to perform the study since there is no explanation on the reproducibility of the observations made and on on how the displayed pictures are representative of the obtained results. Even the description of the quantitative analysis of collagen type I in the materials and methods part does not detail how many samples were used and on how many animals.
The use of antibodies to stain newly formed collagen type I should be properly explained in the materials and methods and results parts, as opposed to be briefly evoked in the discussion part, in order to support the drawn conclusions on collagen synthesis.
In the introduction the clinical interest of such implants should be presented.
In the abstract the number of dogs should be included.
Minor remark: "in the" repeated twice on line 248
Author Response
Dear Reviewer 2
Thank you for the time you have spent on the review. Your insightful, constructive comments are going to improve our manuscript. Any changes in the revised manuscript are in track changes and indicate the location (page and lines) in the new version of the manuscript. Bellow you will find our point-by-point responses.
Reviewer 2: The work deals with the analysis of tissue integration of a porous collagen-elastin scaffold. It is mostly based on immuno-histochemical evaluation and presents original results. Still, major problems need to be adressed before acceptance for publication.
Response to Reviewer 2: Thank you for your careful review.
Reviewer 2: The fabrication of the scaffolds is not detailed, if the reason is industrial secret this should be explained and the main general characteristics should be reported (the composition is only briefly presented in the introduction), including the size of the used scaffolds, which is not even mentioned.
Response to Reviewer 2: Thank you for bringing this to our attention. We have given all information we have about the origin, and chemical and physical properties of the biomaterial on p.2, lines 63-72: “Lately, a collagen-based scaffold with interconnected pores from porcine origin was developed for soft tissue augmentation around teeth and dental implants [7]. Production involves isolating, processing and milling two collagenous tissues, followed by mixing the materials at a 3.5-4.5:0.5-1.5 ratio in an aqueous slurry. After freeze-drying, the resulting composite is treated with a crosslinking agent, washed with water, freeze-dried again, and terminally sterilized by gamma irradiation. The detailed production process is proprietary. The final product has the following characteristics: it is made up of 60-96% (w/w) porcine collagen type I and III and 4-40% elastin and has 93% volume porosity with interconnected pores [8]. Chemical cross-linking was used to increase the stiffness of the scaffold. The average pore diameter is 92 µm.”
This is the complete information that is available for this biomaterial. We have again contacted the company, but they cannot provide details about the production process, since this belongs to the company’s secret.
Regarding the dimensions of the scaffold, we have modified the following sentence on p.2, line 111: “After 90 days, a full thickness flap was elevated and a collagen‐based scaffold (Fibro‐Gide® prototype, Geistlich, Wolhusen, Switzerland), measuring 6 mm x 6 mm x 10 mm, was placed between the bone and the repositioned buccal flap.
Reviewer 2: Along the same line, the authors claim in the discussion that the physical characteristics of biomaterial scaffolds modulate macrophage response and other host reactions, and that TRAP expression is reported for the first time, but no explanation of the obtained results and comparison based on the characteristics of the studied scaffold is proposed.
Response to Reviewer 2: That the physical characteristics of a biomaterial modulate macrophage response and other host reactions is known from other studies. In our experiment, we have used one pre-fabricated scaffold, which was designed from industry. Our goal was not to compare the tissue response, integration and degradation of structural, chemical or physical variants of this biomaterial. TRAP staining is normally associated with osteoclasts, which degrade mineralized bone matrix. To the best of our knowledge, TRAP has not been associated so far with the degradation of a soft tissue biomaterial, as the one investigated here. It is very difficult to make comparison with other biomaterials regarding the TRAP staining, since such data do not exist.
Nevertheless, we have added the following sentence on p.12, line 511-513: “Since the collagenous part of the investigated scaffold underwent degradation, it is very likely that TRAP was involved in this degradation process. Thus far, TRAP was mainly associated with the degradation of the organic bone matrix.”
Reviewer 2: This leads to another flaw of the work, namely the absence of any control to test hypotheses based on the specific features of the studied scaffold. One could imagine that 3 out of the 6 dogs could have undergone implantation with another type of scaffold, or that the implantation could have been asymmetric with the use of different types of scaffold on each side of the maxilla. If these experiments are not possible anymore, the discussion should at least explain why this was not performed and reflect real scientific questioning on the observations made, based on other results from the literature for example.
Response to Reviewer 2: Thank you for this legitimate objection. As a proof of principle, it should first be clarified, if a new biomaterial can principally be used for its intended clinical application. The choice of 6 healing/observation periods was done with the intention to cover many aspects like inflammatory response, invasion of blood vessels, tissue integration and degradation. The 6 healing periods have lowered the number of animals. The control that was done here was a negative control, i.e. sham operation. We have discussed the results of the sham-operated sites in our previous publication. The next step would indeed be a comparison with other biomaterials, but this was not the intention of the present experiment.
We have added the following paragraph at the end of the discussion on p.12, line 525: «Like other studies, the present study has also limitations. One limitation is the small sample size due to the study design, which aimed at studying the sequence of events over a period of 90 days, using 6 observation periods. The study may thus be considered as a pilot experiment and the data as preliminary. Furthermore, only a negative control was used, i.e. sham-operated sites. Future experiments may involve a larger number of tissue samples and other biomaterials for comparative reasons. Nevertheless, the study provides novel and unique data of clinical relevance and for future studies in the biomaterial field.”
Reviewer 2: The results are not quantatively assessed and one might really wonder why 6 dogs were needed to perform the study since there is no explanation on the reproducibility of the observations made and on how the displayed pictures are representative of the obtained results. Even the description of the quantitative analysis of collagen type I in the materials and methods part does not detail how many samples were used and on how many animals.
Response to Reviewer 2: For the whole study, six animals were used because 6 observation periods were chosen in order to describe the changes over time up to 90 days. For the collagen quantification, 3 animals were used. In the Materials & Methods part on p.3, lines 157-216 we explain how the measurements were done: “The area fractions of labeled collagen type I were determined in high-resolution images in 5 rectangles per section and region (i.e. 5 rectangles in the periphery and 5 sections in the center of the collagen-based scaffold). The rectangle size was 0.5 mm x 0.75 mm (area = 0.375 mm2). The central rectangles were placed immediately adjacent to the peripheral rectangles. The rectangles in the periphery were all evenly distributed on the bone side with similar distances between them, the same was true for the central rectangles. The Zeiss Zen software (Zeiss, Oberkochen, Germany) was used to capture the DAB-stained collagen type I and to convert the values into percentages: (labeled area/total area) x 100.”
We have added the following sentence on p.3, line 154-157: “For the quantitative analysis of collagen type I, 3 animals were used (7, 15, and 30 days of healing) because collagen deposition in the scaffold was not observed at 4 hours and at 4 days, and 90 days was not of interest because the active phase of collagen synthesis and deposition was over and replaced by tissue remodeling.”
Reviewer 2: The use of antibodies to stain newly formed collagen type I should be properly explained in the materials and methods and results parts, as opposed to be briefly evoked in the discussion part, in order to support the drawn conclusions on collagen synthesis.
Response to Reviewer 2: The antibody used was specific for canine tissues. For all healing periods, we have checked the labeling for old collagen in the soft connective tissue and in bone, a mineralized connective tissue containing also collagen. The old (not injured) soft connective tissue and the bone matrix were labeled. Labeling was much weaker in these tissues than in the newly formed collagen matrix in the scaffold pores. The negative control incubations showed no labeling at all, neither in the old tissues nor inside the scaffolds.
We have added the following sentence on p7, lines 264-269: “Inspection of the not injured soft connective tissue part of the oral mucosa and of bone revealed positive immunostaining for collagen type I. Immunostaining for collagen type I inside the biomaterial scaffold was first observed after 7 days. Collagen type I labeling progressively increased from the periphery to the central part of the scaffold and the area occupied with labeled collagen type I increased from 7 to 30 days (Figure 4). The labeling intensity was always stronger for the newly formed collagen inside the scaffold than for collagen in the surrounding soft connective tissue and in bone.”
Reviewer 2: In the introduction the clinical interest of such implants should be presented.
Response to Reviewer 2: Thank you for your recommendation. Before the sentence “Recently, a highly porous collagen-based scaffold was developed for soft tissue augmentation around teeth and dental implants” on p.2, lines 61-63, we have added the following sentence: “The autologous connective tissue graft from the palate is still the gold standard for soft tissue augmentations in the oral cavity, but because of additional time and morbidity the search is still going on for the ideal biomaterial to replace the autologous graft.”
Reviewer 2: In the abstract the number of dogs should be included.
Response to Reviewer 2: Thank you. We have changed the sentence on p.1, line 26 into: “After implantation in maxillary submucosal pouches in 6 canines, cell invasion (vimentin), extracellular matrix deposition (collagen type I) and scaffold degradation (cathepsin k, TRAP, CD86) were (immuno)-histochemically evaluated.
Reviewer 2: Minor remark: "in the" repeated twice on line 248
Response to Reviewer 2: Thank you. We have removed the second “in the”.
Reviewer 3 Report
The article is part of the latest research on tissue engineering. the subject of the article is interesting.
The introduction of the article is concise and suggests that the work will concern tissue scaffolds obtained from collagen and elastin. Reading the research methodology, it can be seen that the article focuses on cellular research, the reader must guess where the tissue scaffolding was obtained from. Publication in the Materials journal should contain information about the material being examined, not just about the behavior of the cells with this material. There is no information on how the scaffolding was made, what composition they had, what porosity and strength. Biologically, the article is very good, but it lacks information about the material and its properties. Publication in my opinion needs to be completed.
Author Response
Dear Reviewer 3
Thank you for the time you have spent on the review. Your insightful, constructive comments are going to improve our manuscript. Any changes in the revised manuscript indicate the location (page and lines) in the new version of the manuscript and are highlighted in yellow. Bellow you will find our point-by-point responses.
Reviewer 3: The article is part of the latest research on tissue engineering. the subject of the article is interesting.
Response to Reviewer 3: Thank you for your valuable comments.
Reviewer 3: The introduction of the article is concise and suggests that the work will concern tissue scaffolds obtained from collagen and elastin. Reading the research methodology, it can be seen that the article focuses on cellular research, the reader must guess where the tissue scaffolding was obtained from. Publication in the Materials journal should contain information about the material being examined, not just about the behavior of the cells with this material. There is no information on how the scaffolding was made, what composition they had, what porosity and strength. Biologically, the article is very good, but it lacks information about the material and its properties. Publication in my opinion needs to be completed.
Response to Reviewer 3: Thank you for bringing this to our attention. We have given all information we have about the origin, and chemical and physical properties of the biomaterial on p.2, lines 63-72: “Lately, a collagen-based scaffold with interconnected pores from porcine origin was developed for soft tissue augmentation around teeth and dental implants [7]. Production involves isolating, processing and milling two collagenous tissues, followed by mixing the materials at a 3.5-4.5:0.5-1.5 ratio in an aqueous slurry. After freeze-drying, the resulting composite is treated with a crosslinking agent, washed with water, freeze-dried again, and terminally sterilized by gamma irradiation. The detailed production process is proprietary. The final product has the following characteristics: it is made up of 60-96% (w/w) porcine collagen type I and III and 4-40% elastin and has 93% volume porosity with interconnected pores [8]. Chemical cross-linking was used to increase the stiffness of the scaffold. The average pore diameter is 92 µm.”
This is the complete information that is available for this biomaterial. We have again contacted the company, but they cannot provide details about the production process, since this belongs to the company’s secret.
Round 2
Reviewer 1 Report
Thank you for your reply. Please be aware that I am not questioning or criticizing the scientific merit of this current manuscript per se. The data are indeed scientifically interesting and important, the images are high-standard, the immunos and stainings are beautiful and clear (congrats to whoever performed them on the bench by the way). I also understand the concept of avoiding using other animals but instead trying to generate the most amount of data possible with the same set of samples. I also understand the authors maybe didn't know they would be able to generate the current data in October/November 2019, which in my opinion is a pity. In my opinion, it would make much more sense to have these data in a single paper. My conclusion is that even with the authors' explanation and replies, I do not think the current data has enough novelty to characterize a new paper compared to their previous one (https://www.mdpi.com/1996-1944/12/22/3721), at least not in the same journal (with the same impact factor) of the previous publication. For this to be possible, even not using new animals, I would suggest a much more deep analysis of inflammation, or another aspect of the work, as I have suggested before. Nevertheless, the final decision should be with the editors. Again, my criticism is not on the manuscript per se, but on its conflict of originality compared to previously published manuscript just some months ago.
Author Response
Thank you for your comments and appreciation of our work.
Reviewer 2 Report
Thank you very much for taking into account previous comments. I recommend publication of your work.
Minor remark: inconsistency in the dimensions between what is stated in your response: 6 mm x 6 mm x 10 mm and what is written in the article in its present form line 101.
Author Response
Thank you for your comment. We have corrected this in the manuscript. The correct size is 6 mm x 6 mm x 10 mm (line 104).
Reviewer 3 Report
The article is very interesting, but I still think that there is little information about the material. Thank you for adding information about the pore size, but here you should attach a SEM picture or other result based on which the porosity was calculated, we can not write about the properties without evidence. If the authors publish the results, and not the description of the physico-chemical or mechanical properties of the material, the publication will be very good, without these studies it is a purely biological publication.
Author Response
Thank you for your comments. We have moved the technical information about the scaffold from the introduction to the Material and Methods section as well as expanding the information on the physical properties and how they were analyzed. What we have added is highlighted in yellow and appears in lines 85-96: Production of the collagen-based scaffold (Fibro-Gide® prototype, Geistlich, Wolhusen, Switzerland) involved isolating, processing and milling two collagenous tissues, followed by mixing the materials at a 3.5-4.5:0.5-1.5 ratio in an aqueous slurry. After freeze-drying, the resulting composite was treated with a crosslinking agent, washed with water, freeze-dried again, and terminally sterilized by gamma irradiation. The detailed production process is proprietary. The final product was composed of 60-96% (w/w) porcine collagen type I and III and 4-40% elastin. Furthermore, the average pore diameter was 92 µm and the volume porosity was 93% with interconnected pores, as determined by mercury intrusion porosimetry [8]. Chemical cross-linking was used to increase the stiffness of the scaffold. The critical mechanical properties were described by retention hysteresis under repetitive compressive load of 12.1 kPa, whereby the first order retention was less than 45% after 49 cycles. This equated to a thickness retention after successive loading cycles of more than 85%. The measurements were carried out in saline solution at 37 ºC.